# Nano-Zero-Valent Zinc-Modified Municipal Sludge Biochar for Phosphorus Removal

**DOI:** 10.3390/molecules28073231

**Published:** 2023-04-04

**Authors:** Yupeng Zhang, Wenbo Zhang, Hong Zhang, Dandan He

**Affiliations:** Key Laboratory of Environment-Friendly Composite Materials of the State Ethnic Affairs Commission, Key Laboratory for Utility of Environment-Friendly Composite Materials and Biomass, School of Chemical Engineering, Gansu Provincial Biomass Function Composites Engineering Research Center, University of Gansu Province, Northwest Minzu University, Lanzhou 730030, China

**Keywords:** nano-zero-valent zinc, phosphorus, chemical adsorption, physical adsorption

## Abstract

Municipal sludge biochar (MSBC) can be used to absorb phosphorus in water for waste treatment. Nano-zero-valent zinc (nZVZ) was uniformly attached to MSBC to obtain a highly efficient phosphorus-absorbing composite material, nZVZ–MSBC. Characterization by FTIR, XPS, XRD, and BET showed that nZVZ was uniformly dispersed on the surface of the MSBC. Zinc loading was able to greatly improve the adsorption performance of MSBC for phosphorus. Adsorption experiments illustrated that the adsorption process conformed to the Langmuir model, and the maximum adsorption amount was 186.5 mg/g, which is much higher than that for other municipal sludge biochars. The adsorption process reached 80% of the maximum adsorption capacity at 90 min, and this gradually stabilized after 240 min; adsorption equilibrium was reached within 24 h. The optimum pH for adsorption was 5. The main adsorption mechanism was chemical adsorption, but physical adsorption, external diffusion, internal diffusion, and surface adsorption also played roles. The potential for application as an efficient adsorbent of phosphorus from water was confirmed. In addition, a novel strategy for municipal sludge disposal and resource utilization is provided.

## 1. Introduction

In recent years, the production of municipal sewage has increased rapidly with the gradual acceleration of urbanization. It is predicted that the sludge output of China will reach 90 million tons in 2025 [1]. This would cause secondary pollution without proper disposal and treatment. Secure landfill disposal, incineration, fermentation, and use as building materials are the main ways in which municipal sludge is currently dealt with [2]. However, these disposal methods also have their disadvantages. A secure landfill may cause land pollution and groundwater pollution [3]. The incineration of sludge produces a large amount of harmful waste gas that pollutes the atmospheric environment [4]. Sludge undergoing fermentation releases a large amount of odorous gas, harming the living environment [5]. Furthermore, using sludge to prepare building materials requires a large amount of energy [6]. The preparation of biochar is an alternative treatment method. Municipal sludge contains a large amount of organics and microorganisms, so it is a potential raw material for the preparation of biochar. High-temperature carbonization can not only fully carbonize organic matter but also effectively immobilize microorganisms. The prepared biochar can also be recycled as a resource, especially as an adsorbent for wastewater treatment. In the 1990s, scientists studied the black soil found in the Amazon region (known as terra preta de índio), because of its ability to increase soil fertility and remain stable in the soil for a long time. This was considered the first documented application of biochar [7]. For example, Fan et al. prepared biochar from municipal sludge by pyrolysis at 550 °C and applied it to the adsorption of methylene blue in water. The adsorption capacity of the sample for methylene blue was approximately 20 mg/L [8]. Similarly, Keifer et al. burned municipal sludge at 500 to 700 °C to prepare biochar and used it for the adsorption of ciprofloxacin. The results showed that biochar pyrolyzed at 550 °C has a higher proton-active surface functional group density than that produced at 700 °C, providing more sites for electrostatic interactions with charged aqueous species [9]. Ahmed et al. prepared biochar with a specific surface area that reached 492 m^2^/g and used it for the adsorption of methylene blue in water. Its maximum adsorption capacity was 33 mg/g. [10]. Adsorption is the separation process of transferring guest molecules from the environment to the bulk or surface of the solid or liquid phase [11]. Adsorption and separation technology has been used in various industries [12]. The above research shows that municipal sludge biochar has great potential for treating environmental pollutants.

However, the organic matter content of municipal sludge is low, and the silica content is high. The prepared biochar had a poor pore size, resulting in a low adsorption capacity. We used municipal sludge pyrolyzed at different temperatures to prepare biochar. The adsorption capacity of the biochar was also studied experimentally. We found that biochar samples prepared by using municipal sludge alone had a low adsorption capacity, due to their low pore size. Therefore, it was necessary to modify the municipal sludge biochar to improve its adsorption capacity. Hou et al. found that aluminum-containing sludge had a better adsorption effect for phosphorus. Municipal sludge biochar modified by aluminum nanoparticles had a better phosphorus adsorption capacity, reaching 6.06 mg/g [13]. This inspired researchers to further modify biochar using nanoparticles. Gillingham et al. found that nano-zero-valent iron-modified biochar had greatly improved adsorption capacity for ammonium [14]. Mohamed et al. used nanometer zero-valent zinc to remove acid red dye from water, achieving 90% removal in a few minutes [15]. Hu et al. confirmed that many metal nanoparticles could be loaded onto biochar to improve its adsorption capacity [16]. Using nanoparticles to modify municipal sludge biochar has now become mainstream [17,18]. Based on this, we aimed to remove excessive phosphorus from water using nano-zero-valent metals to address the pollution caused by eutrophication.

In this study, a novel zinc-modified municipal sludge biochar (MSBC) was prepared for the efficient removal of phosphorus from water. Nanometer zero-valent zinc (nZVZ) was added to the composites to increase the number of active sites. To explore the influence of its structural characteristics on its phosphorus adsorption capacity, this study aimed to (1) compare the adsorption capacities of raw biochar and modified biochar to illustrate the enhancement of the phosphorus adsorption capacity; (2) use adsorption kinetics and thermodynamic models to elucidate the synergy occurring in the composites during the adsorption process; and (3) reveal the adsorption mechanism by comprehensive characterization. This study provides an efficient solution for the treatment of phosphine-rich water bodies.

## 2. Results and Discussion

### 2.1. Characterization of the Surface Structure of the Biochar

As shown in Figure 1, FTIR spectra were applied to analyze the surface functional group types of the materials before adsorption and after adsorption. In the range of 4000 to 500, two samples revealed a strength band near 3438 cm^−1^, which was attributed to the –OH stretching vibrations or the sorbed water molecules and water deformation. The deposition of nZVZ on the surface of the biochar was conducive to the tensile vibration of hydroxyl groups. The absorption band around 1632 cm^−1^ was attributed to the vibration of –(OH)_2_^−^, which proved that there was coordination between water molecules and aromatic rings. The bands around 1412 cm^−1^ and 1041 cm^−1^ represented the presence of aliphatic methyl and phenol methyl groups in the biochar, respectively. The peak of Zn-O (720 cm^−1^) was found in the modified material [13]. These zinc oxides might play an important role in the attraction of phosphate at the active adsorption sites. The rich functional groups, after modification, provided a basis for the improvement of the adsorption capacity.

The existence of C, O, Zn, and P elements in nZVZ–MSBC was proven by XPS. The total XPS measurement spectrum (Figure 2a) showed that the material’s prominent element peaks (eV) were C 1s, O 1s, Zn 2p, and P 2p. The high-resolution C 1s spectrum revealed four peaks: C–H (284.4 eV), C–C (285.2 eV), C–O (286.6 eV), and COOH (289.2 eV) (Figure 2b) [19,20]. The binding energy of O 1s had two peaks at 532 eV and 533.4 eV, corresponding to C–O and chemisorbed oxygen (O_C_) (Figure 2c) [21]. However, C–O, with a larger peak area, still dominated the fitting. Figure 2d shows that Zn 2p can be deconvoluted into two parts, Zn 2p_1/2_ (1045.5 eV) and Zn 2p_3/2_ (1022.4 eV) [22]. This was similar to the binding energy of conventional Zn-X-LDX/LDO previously reported (where X represents other metals) [23,24]. Unlike the material before adsorption, the P element appeared in the composite after adsorption. The high-resolution P 2p spectrum in Figure 2e shows three binding types at 133.4 eV, 133.9 eV, and 134.4 eV, respectively, corresponding to P 2p_3/2_, P 2p, and P 2p_1/2_ [20].

Compared with the results before adsorption, the decrease in C–H content and increase in C–C content indicated that chemical reactions were involved in the adsorption process, resulting in a change in chemical bond components. The decrease in the peak area of the O element showed that nano-zero-valent zinc was oxidized in the adsorption process and participated in the reaction. The composition of the Zn element did not change significantly, but the peak area decreased; this may have been related to Zn participating in the reaction and forming a chemical precipitate.

The crystal morphology of the composite was analyzed by X-ray diffraction (XRD), and the data were analyzed using the Jade6.0 software. Figure 3 shows that the diffraction peak of C in the composite was entirely consistent with (PDF # 46-0943), but the diffraction peak of Zn was not detected. This might have been due to the excellent dispersion of nano-zero-valent zinc in the material, which did not form crystals.

According to the results of the BET test, the specific surface area of the nZVZ–MSBC composite reached 162.7 m^2^/g, the total adsorption pore volume was 0.61 cm^3^/g, and the average pore width was 15.1 nm (Table 1). Compared with those of the municipal sludge biochar raw materials, the specific surface area and pore volume more than doubled, and the average pore width slightly increased. This also showed that the nano-zero-valent zinc was uniformly dispersed on the surface of the MSBC and that there was no agglomeration to block the holes [25]. The nitrogen adsorption–desorption curve and pore size distribution curve of the sample are shown in Figure 4, indicating protrusion in the direction of the Y axis in the low-pressure zone and capillary condensation in the high-pressure zone. According to the IUPAC classification, the sample was a type IV isotherm with a typical H3 hysteresis loop. Its adsorption and desorption curves were typical for mesoporous materials. The subsequent pore size distribution curve also proved this point, demonstrating that the material contained abundant mesopores. Meanwhile, the volume of micropores with sizes less than 2 nm was large, which proved that the material also contained a considerable number of micropores, significantly improving the adsorption capacity [26]. This would allow the adsorbed P to occupy the composite’s pores [27]. The curve for MSBC was the lowest, which was also consistent with its average pore size and average pore volume. At the same time, the test results for the pore width and pore volume indicated the sample had decreased significantly after adsorption, further demonstrating this point.

### 2.2. Adsorption Kinetics Study

In Figure 5, the composites’ adsorption capacities for phosphorus at 298 K, 308 K, and 318 K based on the kinetic studies are shown. It can be seen that the adsorption capacity of nZVZ–MSBC for phosphorus rapidly increased in the initial phase over time; 80% of the maximum adsorption capacity was reached in 90 min, and it gradually stabilized within 240 min. The trend of the phosphorus adsorption by the samples under these three temperature conditions was similar, but the maximum adsorption capacity increased with increasing temperature until 186.5 mg/g (at 24 h). The rapid adsorption could be observed in the first 240 min, indicating that phosphorus in the solution was driven to attach to the biochar surface by electrostatic attraction. Ion and ligand exchange controlled the adsorption process and decreased the adsorption rate. It has been reported that biochar produced pyrolytically is not ideal for phosphorus adsorption due to its negative charge [28], and our previous experiments similarly confirmed this. However, the composites showed a much improved phosphorus adsorption capacity after zinc modification. This might have been due to the complexation reaction between phosphorus and the highly active nZVZ, which dramatically improved the adsorption capacity of the sample [29].

To provide insight into the adsorption mechanism, the pseudo-first-order kinetic model, pseudo-second-order kinetic model, and intra-particle diffusion model were fitted to the phosphorus adsorption process. As seen from Figure 5a and Table 2, the pseudo-first-order kinetic model had a low correlation coefficient and thus could not be fitted to this adsorption process (R^2^ < 0.99). The pseudo-second-order kinetic model was better fitted to the present adsorption process, regardless of the temperature conditions, as shown in Figure 5b and Table 2 (R^2^ > 0.99). Its fitting results were more appropriate for the adsorption equilibrium in the actual experiments, indicating that the pseudo-second-order kinetic model was more consistent with the adsorption of phosphorus by the composites. The pseudo-second-order kinetic equation was able to reveal all elements of the adsorption process, including external diffusion, internal diffusion, and surface adsorption, which could be explained as a process of complete phosphorus adsorption by biochar [30]. It also showed that chemical adsorption was the dominant element of the adsorption process.

From Figure 5c, one can see the intraparticle diffusion model could be divided into three stages to describe the adsorption process. The first stage was a rapid surface adsorption control process due to the enrichment of many active sites on the material surface. In the second stage, the adsorption rate decreased. The main reason for this was that, after the surface sites had been consumed, phosphorus began to enter the material’s pores and combine with the functional groups located therein. The third stage was the adsorption saturation stage, where all active sites were occupied and adsorption reached dynamic equilibrium.

### 2.3. Adsorption Thermodynamics Study

We used the Langmuir and Freundlich isotherm models to reveal the process by which the adsorbent adsorbed phosphorus at different concentrations and temperatures, as shown in Figure 6a,b. It was obvious that the adsorption capacity increased with increasing temperature, which indicated that the reaction was endothermic. We found that the amount adsorbed was positively related to the concentration of phosphorus in the adsorbent solution, and adsorption saturation was reached within 24 h. The maximum adsorption capacity was reached faster when the concentration of the phosphorus solution was lower. This may indicate that all the adsorption sites of the composite were gradually occupied under a limited concentration, and the maximum adsorption capacity was approximately 312.99 mg/g.

The relevant calculation parameters for the two models are shown in Table 3. The Langmuir model showed a better fit for the natural adsorption process under three different temperature conditions (R^2^ > 0.99). Some studies have shown that the adsorption process is dominated by single-layer homogeneous adsorption [31] and that nZVZ adheres evenly to the surface of MSBC. The highly active nZVZ was uniformly attached to the surface of MSBC in this study, which could have supported the complexation reaction between phosphorus and the adsorbent in the physical adsorption process, improving the adsorption capacity of the biochar.

Figure 6c shows the fitting graph for the Sips model. The Sips model is a combination of the Langmuir and Freundlich models. The Sips model’s constant m reflects the adsorption process’s heterogeneity. When the value of m falls below 1, the model is inclined toward Langmuir fitting; when it rises above 1, the model is biased toward Freundlich fitting. It can be seen from Table 3 that the correlation of the Sips model was high (R^2^ > 0.99), indicating that the Sips model fit the adsorption process well. At the same time, the m value was between 0.6649 and 0.6826, indicating that the Langmuir model better reflected the adsorption process, which was consistent with the previous analysis results.

### 2.4. Effect of pH on Phosphorus Adsorption Capacity

Batch adsorption experiments were conducted in which the pH was fixed at different values, from 2 to 12, as the only variable (Figure 7). As the pH was increased from 2 to 5, the adsorption capacity gradually increased, peaked at pH 5, and then decreased gradually as the pH was further increased until the pH was 12.

The zero point of charge of the biochar was between 5 and 7; when the pH of the solution was lower than 5, a hydroxylation reaction occurred on the surface of the biochar, and a colloidal interface X-OH^2+^ was formed (where X represents the active center) [32]. At this point in time, the biochar surface was positively charged and reacted with the negatively charged phosphate radicals through electrostatic attraction, removing phosphorus from the solution. When the pH of the solution was greater than 7, the biochar surface had a negative charge, causing it to react with phosphate radicals through electrostatic repulsion, resulting in a gradual reduction in the phosphorus adsorption capacity of the biochar.

On the other hand, at different pH values, the existing forms of phosphorus were also different. The ionization equilibrium constants corresponding to $H_{2}{\mathrm{PO}}_{4}^{-}$, ${\mathrm{HPO}}_{4}^{2-}$, and ${\mathrm{PO}}_{4}^{3-}$ were 2.15, 7.20, and 12.33, respectively [33]. When the pH was between 2.15 and 7.2, the $H_{2}{\mathrm{PO}}_{4}^{-}$ content increased with the pH of the solution. At this point, the adsorption of phosphorus by the biochar mainly depended on a physical action: the electrostatic attraction between the positive charge on the surface of the biochar and the phosphate radicals. Therefore, when the pH of the solution increased, the rate at which the biochar removed phosphorus also increased. However, when the pH rose from 7.2 to 12.33, phosphorus mainly existed in the form of ${\mathrm{HPO}}_{4}^{2-}$, and the content of OH^−^ increased with the pH of the solution. At this point, the hydroxyl radicals in the solution also competed with the phosphate radicals for adsorption, consuming the active sites on the surface of the biochar and resulting in a gradual reduction in the rate at which the biochar removed phosphorus [34,35,36]. In addition, at a high pH, the form of phosphate meant that more adsorption sites on the biochar were needed, which also led to a decrease in the adsorption capacity [37].

## 3. Materials and Methods

### 3.1. Materials and Reagents

Municipal sludge was collected from a sewage treatment plant in Yuzhong, Lanzhou, Gansu, China. The sample was dehydrated, dried, ground, and sieved through a 100-mesh screen. All the chemicals used in this study were of analytical grade. Sodium borohydride (NaBH_4_), zinc chloride (ZnCl_2_), and potassium phosphate monobasic (KH_2_PO_4_) were purchased from Tianjin Damao Chemicals Co. Ltd., Tianjin, China.

### 3.2. Biochar Preparation

The raw materials were loaded into the tubular furnace using a corundum boat and pyrolyzed at 350 °C; the reaction was carried out under a nitrogen atmosphere for 1 h. The material was then oven-dried at 60 °C after multiple washes with deionized water. We mixed half mass KOH and ground it, pyrolyzed it at 800 °C in a tubular furnace for 2 h under a nitrogen atmosphere, immersed it in 1 mol/L HCl for 12 h to remove the ash and excess KOH, centrifuged and rinsed it with deionized water until the pH was stable, and then dried it, acquiring municipal sludge biochar (MSBC). A total of 1 g of MSBC and 12 mmol of ZnCl_2_ (1.6356 g) were mixed into 50 mL of deionized water by stirring for 12 h. Then, 50 mL of NaBH_4_ (0.681 g/50 mL) was added dropwise, and the mixture was centrifuged, vacuumed, and freeze-dried to prepare the nZVZ–MSBC.

### 3.3. Biochar Characterization

The crystal structures of different biochar preparations were studied by X-ray diffraction (XRD, XRD, PANalytical, Almeno, Netherlands). The functional groups on the surface of the biochar were identified using Fourier transform infrared spectroscopy (FTIR, VERTEX 70, Bruker, Germany). The Brunauer–Emmett–Teller (BET) specific surface area was determined using a nitrogen adsorption-specific surface tester, ASAP2020M (Micromerics Instrument Corporation, Norcross, GA, USA), and the specific surface area and pore size distribution curve of the porous materials were determined by the nitrogen low-temperature adsorption method. X-ray photoelectron spectroscopy (XPS) was conducted using a Thermo Scientific ESCALAB250Xi with a monochromatic Al Kα X-ray source (hν = 169 1486.68 eV) at a take-off angle of 90°. All images are drawn using Origin 2022b software developed by OriginLab in the United States.

### 3.4. Batch Adsorption Experiments

A series of batch adsorption experiments were performed in a thermostatic water bath shaker oscillating at 200 rpm at 25, 35, and 45 °C to detect the adsorption behavior and mechanism of the nZVZ–MSBC for total phosphorus. KH_2_PO_4_ was dissolved separately in deionized water, and we used an ultraviolet spectrophotometer to measure the phosphorus concentration of the adsorbed solution by molybdenum antimony spectrophotometry at a 700 nm wavelength. Three parallel experiments were conducted for all the experiments, and the average values were taken as the experimental results. The adsorption capacity at different times (q_t_, mg/g) and the equilibrium adsorption capacity (q_e_, mg/g) of the biochar were defined according to the following equations:(1)$$q_{t}=\frac{{(C}_{0}{-C}_{t})/V}{M}$$
(2)$$q_{e}=\frac{{(C}_{0}{-C}_{e})/V}{M}$$
where C_0_, C_t_, and C_e_ represent the initial concentration, the concentration at the end of the reaction time, and the equilibrium concentration of phosphorus (mg/L), respectively, and V is the volume of the reaction solution (mL), while M stands for the biochar mass (mg).

#### 3.4.1. Adsorption Kinetics Experiment

We added 50 mL of a 100 mg/L KH_2_PO_4_ solution into conical flasks containing 25 mg of nZVZ–MSBC, placed the flasks into a thermostatic water bath shaker, and oscillated them. The mixtures were removed after 5, 15, 30, 45, 60, 90, 120, 240, 480, 720, 960, and 1440 min. The experimental results were fitted with the following three adsorption kinetic models:(3)$$\mathrm{Pseudo}\text{-}\mathrm{first}\text{-}\mathrm{order}\;\mathrm{kinetic}:\;q_{t}{=q}_{e}\left(1-e^{-k_{1}t}\right)$$
(4)$$\mathrm{Pseudo}\text{-}\mathrm{second}\text{-}\mathrm{order}\;\mathrm{kinetic}:\;q_{t}=\frac{k_{2}q_{e}^{2}t}{{1+k}_{2}q_{e}t}$$
(5)$$\mathrm{Intra}\text{-}\mathrm{particle}\;\mathrm{diffusion}\;\mathrm{model}:\;q_{t}{=k}_{3}t^{0.5}+C$$
where q_e_ (mg/g) and q_t_ (mg/g) are the adsorption amount at equilibrium and the adsorption amount homologous to the reaction time t (min) for ${\mathrm{PO}}_{4}^{3-}$; k_1_ (min^−1^), k_2_ (g·mg^−1^·min^−1^), and k_3_ (mg·g^−1^·min^−0.5^) are the respective rate constants of the three models; and C (mg/g) is the desorption constant of the intra-particle diffusion model.

#### 3.4.2. Isothermal Adsorption Experiment

We evaluated the equilibrium isotherm by changing the initial concentration of ${\mathrm{PO}}_{4}^{3-}$; adding 50, 100, 200, 300, 500, 800, and 1000 mg/L of KH_2_PO_4_ solution using the same experimental method as above; and then oscillating the samples in a constant temperature water bath oscillator for 24 h to achieve adsorption equilibrium before calculating the adsorption amount. Langmuir, Freundlich, and Sips models were selected to fit the experimental results:(6)$$\mathrm{Langmuir}\;\mathrm{model}:\;Q_{e}=\frac{C_{e}Q_{m}K_{L}}{{1+C}_{e}K_{L}}$$
(7)$$\mathrm{Freundlich}\;\mathrm{model}:\;Q_{e}{=K}_{F}C_{e}^{1/n}$$
(8)$$\mathrm{Sips}\;\mathrm{model}:\;Qe_{\;}=\frac{Q_{m}{{(K}_{S}C_{e})}^{m}}{{1+(K}_{S}C_{e}{)}^{m}}$$
where Q_e_ (mg/g) and C_e_ (mg/L) refer to the amount of ${\mathrm{PO}}_{4}^{3-}$ adsorbed at equilibrium and the concentration of ${\mathrm{PO}}_{4}^{3-}$ in the solution, respectively; Q_m_ (mg/g) represents the maximum amount of ${\mathrm{PO}}_{4}^{3-}$ adsorbed; K_L_ is the adsorption affinity constant; K_F_ stands for the Freundlich constant; 1/n is the non-linear constant; and K_S_ and m are Sips constants.

## 4. Conclusions

In this study, a novel biochar adsorbent was successfully prepared using nanometer zero-valent zinc for the modification of municipal sludge biochar, and efficiently employed for the removal of phosphorus from aqueous solution. The kinetic results showed that the composite material could absorb most of the phosphorus in the water within 90 min. The maximum adsorption capacity was 186.5 mg/g. The isotherm study showed that, when the initial phosphorus concentration was 1000 mg/L, the adsorption capacity of the composite reached 312.99 mg/g. This revealed that the adsorbent can not only rapidly adsorb a large amount of phosphorus in a short period of time but also that the maximum adsorption amount is considerable. The combination of the highly active nZVZ and the MSBC’s large specific surface area improved the adsorption performance significantly. The batch experiments illustrated that the adsorption process fitted the pseudo-second-order model and Langmuir model. The adsorption process was dominated by chemisorption, including external diffusion, internal diffusion, and surface adsorption. At the same time, the occurrence of complexation reactions and the formation of new valences also played a significant role in the adsorption of composite materials. The results demonstrated a practical and efficient method for the utilization of municipal sludge as a resource. The nZVZ–MSBC may have potential for application in the treatment of phosphorus-rich wastewater.

## Figures and Tables

**Figure 1 molecules-28-03231-f001:**
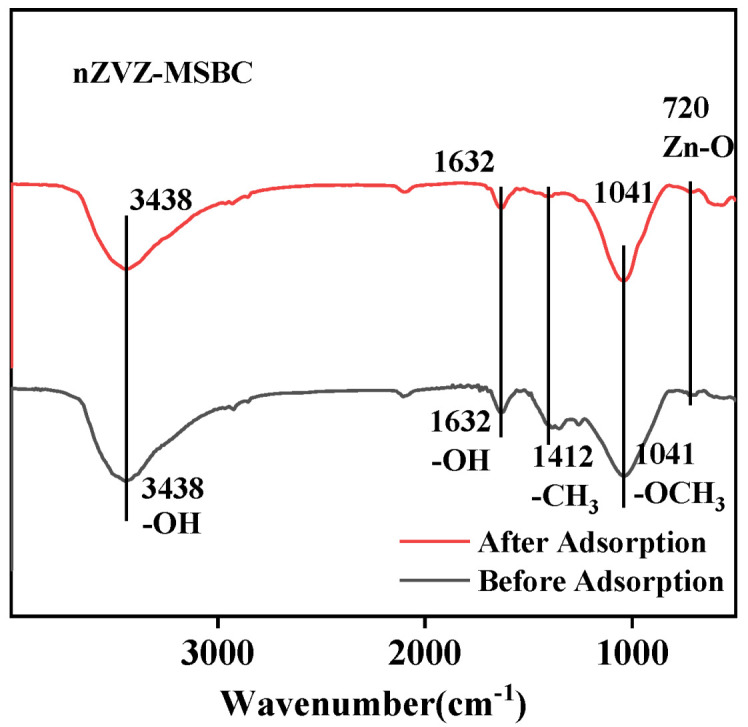
FTIR results before and after adsorption of nZVZ–MSBC.

**Figure 2 molecules-28-03231-f002:**
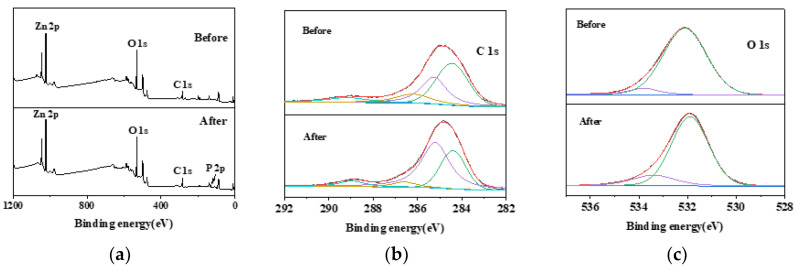
XPS results before and after adsorption of nZVZ–MSBC. (**a**) Total sample spectrum, (**b**) C 1s spectrum, (**c**) O 1s spectrum, (**d**) Zn 2p spectrum, and (**e**) P 2p spectrum.

**Figure 3 molecules-28-03231-f003:**
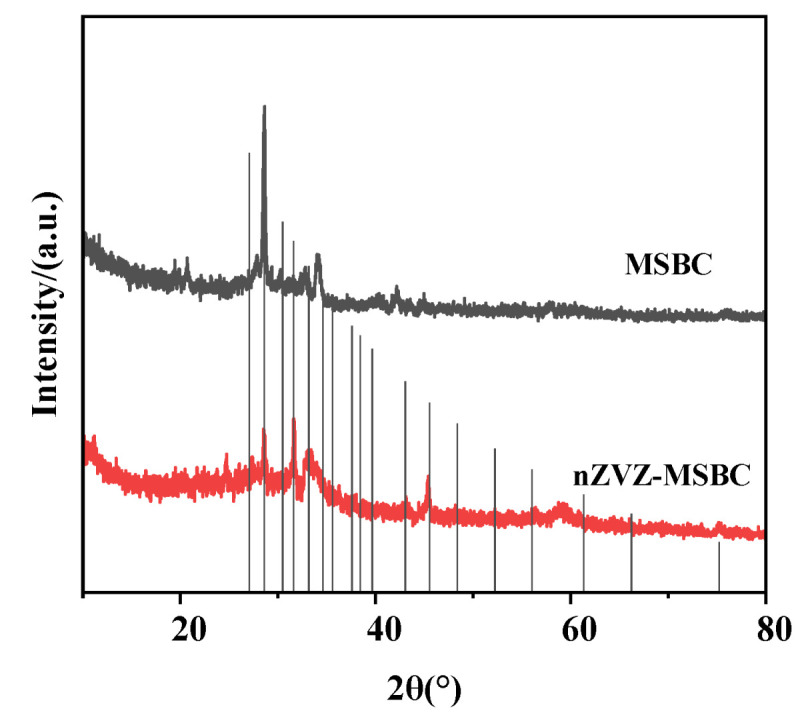
XRD results of MSBC and nZVZ–MSBC.

**Figure 4 molecules-28-03231-f004:**
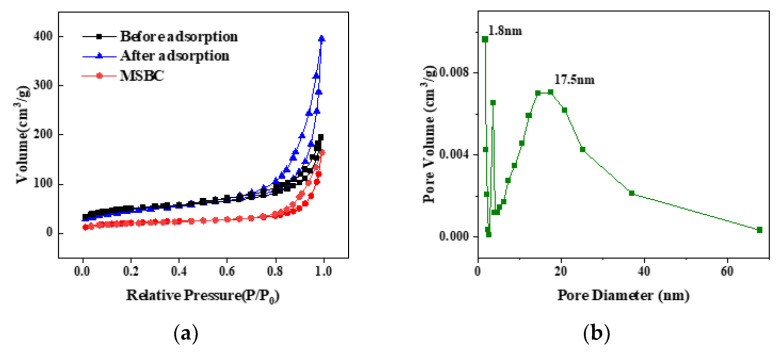
BET results for nZVZ–MSBC. (**a**) Nitrogen adsorption–desorption curve before and after adsorption and (**b**) pore size distribution curve.

**Figure 5 molecules-28-03231-f005:**
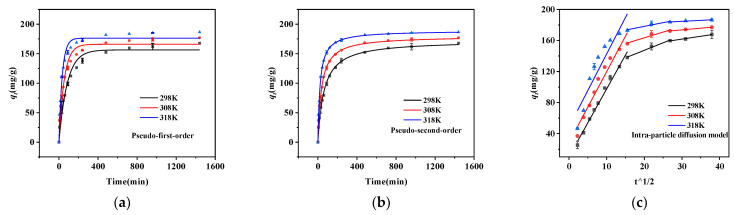
Adsorption kinetics study. (**a**) Pseudo-first-order model, (**b**) pseudo-second-order model, and (**c**) intra-particle diffusion model.

**Figure 6 molecules-28-03231-f006:**
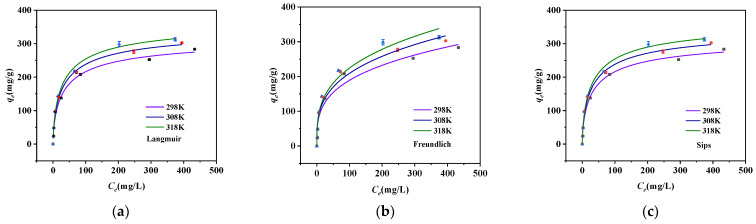
Adsorption thermodynamics study. (**a**) Langmuir model, (**b**) Freundlich model, and (**c**) Sips model.

**Figure 7 molecules-28-03231-f007:**
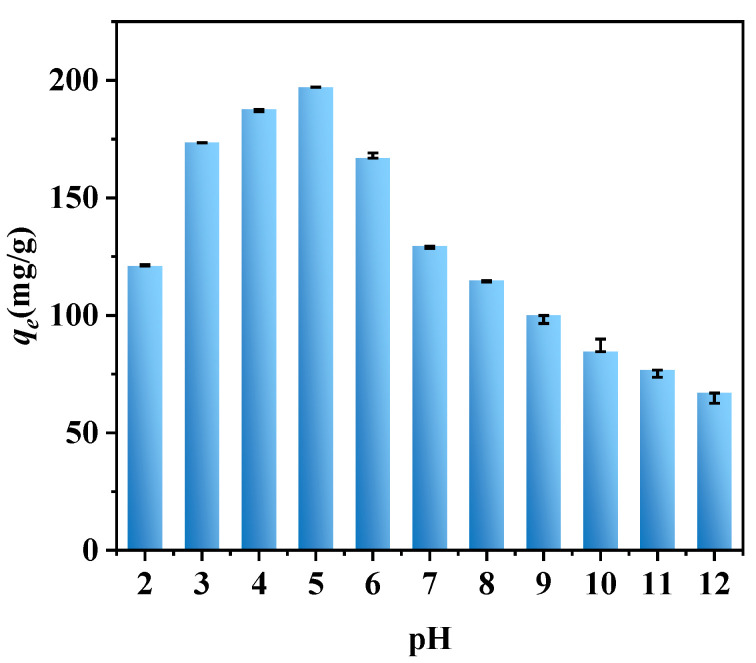
Effect of different pH levels on the phosphorus adsorption capacity.

**Table 1 molecules-28-03231-t001:** Characteristics of the pore structures of adsorption materials.

Material	MSBC	nZVZ–MSBC	nZVZ–MSBC (After Adsorption)
S_BET_ (m^2^/g)	71.24	162.77	166.99
D_pore_ (nm)	14.30	15.05	7.21
V_pore_ (cm^3^/g)	0.25	0.61	0.30

**Table 2 molecules-28-03231-t002:** Adsorption kinetic parameters of the biochar.

Material	nZVZ–MSBC		
T (K)	298	308	318
q_e_ ^a^	167.77	177.04	186.51
Pseudo-first-order			
q_e_ ^b^ (mg/g)	156.42	165.94	176.37
k_1_ (min)	0.01185	0.01859	0.02965
R^2^	0.9704	0.9613	0.97161
Pseudo-second-order			
q_e_ ^b^ (mg/g)	172.55	179.88	189.16
k_2_ g/(mg·min)	9.29 × 10^−5^	1.52 × 10^−4^	2.43 × 10^−4^
R^2^	0.99293	0.99061	0.99581

^a^ Experimental values. ^b^ Calculated values.

**Table 3 molecules-28-03231-t003:** Thermodynamic parameters of the biochar.

T (K)	298	308	318
Langmuir			
K_L_ (L/mg)	0.0901	0.0924	0.0896
Q_m_ (mg/g)	328.51	354.67	376.41
R^2^	0.99045	0.99465	0.99237
Freundlich			
K_F_ (mg/g(L/mg)1/n)	48.395	51.800	53.770
1/n	0.03299	0.03164	0.03478
R^2^	0.9684	0.97186	0.96727
Sips			
Q_m_ (mg/g)	328.58	354.75	376.54
m	0.6649	0.67283	0.6826
K_S_ (L/mg)	0.02679	0.02902	0.02921
R^2^	0.99279	0.99589	0.99407

## Data Availability

Research data not available in the manuscript can be obtained from the corresponding author upon reasonable request via email.

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
