# Peer review of "Nano-Zero-Valent Zinc-Modified Municipal Sludge Biochar for Phosphorus Removal"

_molecules, 2023, doi:10.3390/molecules28073231_

Round 1

Reviewer 1 Report

The manuscript named Nano zero-valent zinc modified municipal sludge biochar for phosphorus removal is an interesting topic, especially since the amount of sludge is getting bigger and bigger. The article claims a novel strategy of municipal sludge disposal and resource utilization. Yet I do not feel I have read strategy for disposal and resource utilization, neither this is a novel strategy. In the introduction, the authors themselves present studies that have already done something like this. I think this claim is very inappropriate, the article should add some strategy about it.

In the introduction the authors said... This study provided a cost-effective solution for the phosphine-rich treatment of water bodies.. although I didn't read in the paper any cost analysis?

The section 2.5. It doesn't make sense for this part to stand there

The writing needs to be improved. The logic of the manuscript is hard to follow and there are many typos.

Author Response

Response to Reviewer 1 Comments

    Thank you very much for your valuable comments. First of all, I want to apologize for my grammatical mistakes. We have used MDPI's English editing service to modify the syntax of the manuscript. The red and blue marks indicate the modifications made to the syntax. The green mark is a modification we made to your comments.

Point 1: The manuscript named Nano zero-valent zinc modified municipal sludge biochar for phosphorus removal is an interesting topic, especially since the amount of sludge is getting bigger and bigger. The article claims a novel strategy of municipal sludge disposal and resource utilization. Yet I do not feel I have read strategy for disposal and resource utilization, neither this is a novel strategy. In the introduction, the authors themselves present studies that have already done something like this. I think this claim is very inappropriate, the article should add some strategy about it.

Response 1: Thank you for your suggestion, which is very valuable for our article. We added some strategies related to our previous experiments using the green modification markers on lines 67 to 71.(“We used municipal sludge pyrolyzed at different temperatures to prepare biochar. The adsorption capacity of the biochar was also studied experimentally. We found that biochar samples prepared by using municipal sludge alone had a low adsorption capacity, due to their low pore. Therefore, it was necessary to modify the municipal sludge biochar to improve its adsorption capacity.”)

Point 2: In the introduction the authors said... This study provided a cost-effective solution for the phosphine-rich treatment of water bodies.. although I didn't read in the paper any cost analysis?

Response 2: Thank you very much for your comments. I'm sorry for my English expression problem, but the word I wanted to use was effective rather than cost. I'm very sorry that I used the wrong vocabulary to add trouble to your reading. We use the green flag to change cost-effective to effective on line 94.

Point 3: The section 2.5. It doesn't make sense for this part to stand there.

Response 3: Thank you for this comment. We attach great importance to this comment. After re reading the entire article, we reached the same conclusion as you. So we deleted this section in The section 2.5.

Point 4: The writing needs to be improved. The logic of the manuscript is hard to follow and there are many typos.

Response 4: Thank you very much for your comments. I am very sorry for the inconvenience caused to your reading. We have used MDPI's English editing service and have modified the syntax of the article with red and blue markings. I hope the revised manuscript will read more smoothly.

Reviewer 2 Report

The authors have submitted an intriguing article entitled " Nano zero-valent zinc modified municipal sludge biochar for phosphorus removal" which deals with the synthesis, characterization, and application of a novel zinc-modified municipal sludge biochar for efficient phosphorus removal from water. After careful review, I have decided that this paper has the potential to make a valuable contribution to the field. However, I kindly request that the authors revise the manuscript considering the following suggestions. Finally, I propose that this article be published after undergoing a rigorous revision.

General Comments:

1- The abstract is explicit and succinct, encapsulating all the salient aspects, including a brief/general introduction to the subject matter, a non-technical summary of the major findings, and their implications. However, it can be improved.

2- The introduction is engaging, unambiguous, and succinct. The introductory segment provides a comprehensive portrayal of the challenge/gap, underpinned by an incisive and thorough literature review, but it could benefit from further improvement.

3- The various segments of the main body are unambiguous and concise in their entirety.

4- The experimental design and characterization section is cogent; however, some comments need addressing, and some concerns require attention.

5- The conclusions are well-grounded.

Suggested revisions:

1- First and foremost, I strongly urge the authors to provide an enhanced (or straightforward, high-quality, and informative) "Graphical abstract," which can convey the entire concept of their study at a glance. I would suggest that the authors devise a "Graphical Abstract" for this study to portray the entire story in an uncomplicated and informative manner.

2- The abstract should be revised in a way that includes the most important empirical results of the research. These empirical results are the foundation of the research, and they are what make the study relevant and significant. Therefore, including them in the abstract is crucial to give readers an immediate understanding of the study's findings.

3- Kindly scrutinize the manuscript meticulously to eliminate any grammatical errors, typos, and vague sentences. Some of the sentences are needlessly long, rendering them dreary and perplexing for readers to comprehend. Please ensure that the entire manuscript is double-checked and revised in its entirety.

4- The novelty statement of an article is of immense significance as it highlights the importance of the current study and sets it apart from previously conducted research. In this study, the novelty statement inadequately represents the work, and the authors require more development to refine their hypothesis and objectives and elucidate how the presented work differs from other previously and recently published reports in the field.

5- Given the wealth of recent research in this area, the authors must update their references and include more recent studies to provide a more comprehensive overview of the field. Additionally, it is recommended that the authors draw on the following key publications to enhance their discussion of the fundamental concepts in this field:

Introduction section/ page 2/lines 39-45: https://doi.org/10.1016/B978-0-12-818805-7.00001-1

Introduction section/ page 3/ line 43: https://doi.org/10.1016/B978-0-12-818805-7.00009-6

6- To increase the validity of the data obtained, it is highly recommended that the researchers of this study repeat all experiments at least three times with appropriate statistical analyses, such as those depicted in Figure 7 and other comparable results. This measure enables researchers to maintain a level of confidence in the authenticity and reliability of their findings, thus reducing the likelihood of chance errors.

Author Response

Response to Reviewer 2 Comments

    Thank you very much for your valuable comments. First of all, I want to apologize for my grammatical mistakes. We have used MDPI's English editing service to modify the syntax of the manuscript. The red and blue marks indicate the modifications made to the syntax. The green mark is a modification we made to your comments.

Point 1: First and foremost, I strongly urge the authors to provide an enhanced (or straightforward, high-quality, and informative) "Graphical abstract," which can convey the entire concept of their study at a glance. I would suggest that the authors devise a "Graphical Abstract" for this study to portray the entire story in an uncomplicated and informative manner.

Response 1: Thank you for your suggestion, which is very valuable for our article. We re-edited the Graphical Abstract to describe the entire article. The following image is our Graphical Abstract.

Graphical abstract

Point 2: The abstract should be revised in a way that includes the most important empirical results of the research. These empirical results are the foundation of the research, and they are what make the study relevant and significant. Therefore, including them in the abstract is crucial to give readers an immediate understanding of the study's findings.

Response 2: Thank you very much for your comments. We have added more experimental results using green markers on lines 18 to 20 of the summary. Help readers quickly understand our research.

Point 3: Kindly scrutinize the manuscript meticulously to eliminate any grammatical errors, typos, and vague sentences. Some of the sentences are needlessly long, rendering them dreary and perplexing for readers to comprehend. Please ensure that the entire manuscript is double-checked and revised in its entirety.

Response 3: Thank you for this comment. I'm very sorry because my English level is not good, which has added difficulties to readers' reading. We have used MDPI's English editing service to modify the entire article. And mark the modified sentences with red and blue markers. I hope the revised manuscript will make it easier for readers to read.

Point 4: The novelty statement of an article is of immense significance as it highlights the importance of the current study and sets it apart from previously conducted research. In this study, the novelty statement inadequately represents the work, and the authors require more development to refine their hypothesis and objectives and elucidate how the presented work differs from other previously and recently published reports in the field.

Response 4: Thank you very much for your comments, which are of great significance to our article. We compare our plan with other plans on lines 33 to 40. We describe other people's research in more detail on lines 52 to 60. All revisions to the manuscript above are marked with green marks. I hope these changes can show how our research differs from others.

Point 5: Given the wealth of recent research in this area, the authors must update their references and include more recent studies to provide a more comprehensive overview of the field. Additionally, it is recommended that the authors draw on the following key publications to enhance their discussion of the fundamental concepts in this field:

Introduction section/ page 2/lines 39-45: https://doi.org/10.1016/B978-0-12-818805-7.00001-1

Introduction section/ page 3/ line 43: https://doi.org/10.1016/B978-0-12-818805-7.00009-6

Response 5: Thank you very much for your recommended publication. We read carefully and have been very helpful to our article. We refer to these two publications on lines 60 to 62. And use them as references 11 and 12.

Point 6: To increase the validity of the data obtained, it is highly recommended that the researchers of this study repeat all experiments at least three times with appropriate statistical analyses, such as those depicted in Figure 7 and other comparable results. This measure enables researchers to maintain a level of confidence in the authenticity and reliability of their findings, thus reducing the likelihood of chance errors.

Response 6: Thank you very much for your comments. As we have explained in the article, all experiments were repeated three times and the final data were averaged(Lines 335 to 336). To show the authenticity of the experimental data, we have added error bars to all the data graphs and updated the graphs.

Round 2

Reviewer 2 Report

The manuscript is well-amended and I have no further comments.